**Subject Area:**
biochemistry/structural biology

RIO kinase, RIOK2, inhibitor, ribosome assembly, 40s, crystal structure

**Author for correspondence:**
Jonathan M. Elkins
e-mail: jon.elkins@sgc.ox.ac.uk

# Crystal structure of human RIOK2 bound to a specific inhibitor

Jing Wang[1], Thibault Varin[2], Michal Vieth[3] and Jonathan M. Elkins[1,4]

[1]Structural Genomics Consortium, Nuffield Department of Clinical Medicine, University of Oxford, Old Road Campus Research Building, Roosevelt Drive, Oxford OX3 7DQ, UK
[2]Discovery Chemistry Research and Technologies, Eli Lilly and Company, Lilly Corporate Center, Indianapolis, IN 46285, USA
[3]Discovery Chemistry Research and Technologies, Eli Lilly and Company, Lilly Biotechnology Center, 10290 Campus Point Drive, San Diego, CA 92121, USA
[4]Structural Genomics Consortium, Universidade Estadual de Campinas, Cidade Universitária Zeferino Vaz, Av. Dr. André Tosello 550, Barão Geraldo, Campinas/SP 13083-886, Brazil

JME, 0000-0003-2858-8929

The RIO kinases (RIOKs) are a universal family of atypical kinases that are essential for assembly of the pre-40S ribosome complex. Here, we present the crystal structure of human RIO kinase 2 (RIOK2) bound to a specific inhibitor. This first crystal structure of an inhibitor-bound RIO kinase reveals the binding mode of the inhibitor and explains the structure–activity relationship of the inhibitor series. The inhibitor binds in the ATP-binding site and forms extensive hydrophobic interactions with residues at the entrance to the ATP-binding site. Analysis of the conservation of active site residues reveals the reasons for the specificity of the inhibitor for RIOK2 over RIOK1 and RIOK3, and it provides a template for inhibitor design against the human RIOK family.

## 1. Introduction

The RIO kinases (RIOKs) are an evolutionarily conserved family of atypical kinases that are present in all eukaryotes, most archaea and some prokaryotes [1–4]. The RIOKs have the protein kinase fold but lack the sequence motifs involved in substrate binding, including the activation loop. Their presence in prokaryotes suggests that they are the original enzymes with the protein kinase fold, and that protein kinases capable of phosphorylating substrate proteins were a later development [1].

RIOK1 and RIOK2 are found in all eukaryotes, while higher eukaryotes have an additional RIOK, RIOK3. All three RIOKs are found in the pre-40S ribosome, although involved at different stages of the maturation process. Ribosome assembly is a lengthy process with many regulatory steps and includes disassembly of some components followed by further assembly [5,6]. The deficiency of RIOK1 or RIOK2 prevents the formation of mature 40S ribosomes, and in yeast both RIOK1 and RIOK2 are essential for cytoplasmic maturation steps starting from the 20S pre-rRNA [7,8]. It appears that RIOK1 and RIOK2 have distinct roles; RIOK2 associates and then dissociates from the maturing pre-40S ribosome before RIOK1 binds to a very late pre-40S ribosome [9].

Since it appears that the RIO kinases lack the surface regions for substrate binding present in other eukaryotic protein kinases, the evidence suggests that the major role of RIOKs is as an ATPase [10]. RIOK1 and RIOK2 do have auto-phosphorylation activity on an active site aspartate residue, and it is likely that this plays a regulatory role [9,10].

RIOK2 is highly expressed in a variety of cancers [11,12], including non-small-cell lung cancer where high RIOK2 expression was correlated with poor outcome [13]. Progression through the cell cycle is dependent upon

translation capacity in the cell, and whether these high expression levels are simply correlated to increased rates of ribosome assembly necessary for protein synthesis, or whether RIOK2 can acquire additional oncogenic functions not linked to ribosome assembly has not been studied. There is some evidence for the latter from a *Drosophila* model of glioblastoma where RIOK1 and RIOK2 were shown to drive proliferation and survival, and where RIOK1 and RIOK2 expressions were linked to oncogenic AKT signalling [14]. RIOK1 and RIOK2 may have signalling roles downstream of phosphoinositide 3-kinases (PI3Ks) and receptor tyrosine kinases including EGFR, possibly involving the TORC2 complex [14]. Multiple phosphorylation sites have been found on RIOK2, most of which are located on the C-terminal part of the protein located after the kinase domain [15]. Several of these sites are phosphorylated by PLK1 and there appears to be a mechanistic link between PLK1 phosphorylation of RIOK2 and mitosis, with PLK1 phosphorylation of RIOK2 regulating metaphase to anaphase transition [16].

Structures of RIOK2 from two different thermophilic organisms have been reported: the archaea *Archaeoglobus fulgidus* [17] and the fungus *Chaetomium thermophilum* [10]. Furthermore, the cryo-electron microscopy (cryo-EM) structure of a yeast late cytoplasmic 40S ribosome was obtained using the material purified using TAP-tagged Rio2 [18], and revealed some of the binding interfaces of Rio2 within the maturing 40S ribosome. Additional efforts to examine pre-40S complexes from yeast or human by cryo-EM have shown that there is considerable conformational heterogeneity in the binding of RIOK2 that varies with the assembly stage of the pre-40S complex [19,20]. There are also crystal structures of RIOK1 from *A. fulgidus* [21] and human [9].

Here, we report the X-ray crystal structure of human RIOK2 and analyse it in relation to previously known RIOK structures. The distribution of conserved residues demonstrates the importance of ATP binding and hydrolysis for the function of RIOK2. In addition, as we have co-crystallized human RIOK2 with a selective RIOK2 inhibitor [22], we are able to analyse the structural basis for inhibitor selectivity. As the first inhibitor-bound RIOK2 structure, these data support the future design of small molecules that could be used to identify non-ribosome-assembly functions of RIOK2 as well as block the catalytic activity of RIOK2 when bound in the pre-40S ribosome complex, to allow understanding of the role of ATP hydrolysis and RIOK2 conformational change in ribosome assembly.

## 2. Results

### 2.1. Structure determination of RIOK2

Since the kinase domain of RIOK2 is at the N-terminus, we constructed various different C-terminal truncations of RIOK2 in *Escherichia coli* expression plasmids and assessed each of them for their ability to overexpress soluble RIOK2 kinase domain. Several different truncations of RIOK2 expressed good yields of soluble protein when combined as a fusion protein with the Zbasic protein domain [23]. After partial protein purification, the Zbasic domain could be cleaved from RIOK2 yielding stable and soluble RIOK2 kinase domain protein. These RIOK2 proteins were used in

**Table 1.** Data collection and refinement statistics.

royalsocietypublishing.org/journal/rsob    Open Biol. 9: 190037

| | RIOK2 : compound 9 |
|---|---|
| PDB ID | 6HK6 |
| space group | $P2_1$ |
| no. of molecules in the asymmetric unit | 10 |
| unit cell dimensions *a*, *b*, *c* (Å), | 92.0, 92.9, 237.7, |
| $\alpha$, $\beta$, $\gamma$ (°) | 90, 99.1, 90 |
| *data collection* | |
| resolution range (Å)[a] | 92.86 – 2.35 (2.39 – 2.35) |
| unique observations[a] | 164 333 (8004) |
| average multiplicity[a] | 3.1 (2.8) |
| completeness (%)[a] | 99.7 (98.3) |
| $R_{merge}$[a] | 0.11 (0.59) |
| mean $((I)/\sigma(I))$[a] | 6.6 (1.8) |
| mean CC(1/2) | 0.991 (0.482) |
| *refinement* | |
| resolution range (Å) | 78.34 – 2.35 |
| *R*-value, $R_{free}$ | 0.22, 0.24 |
| r.m.s. deviation from ideal bond length (Å) | 0.008 |
| r.m.s. deviation from ideal bond angle (°) | 1.42 |
| Ramachandran outliers | 0.00% |
| most favoured | 97.95% |

[a]Values within parentheses refer to the highest-resolution shell.

co-crystallization trials with the published specific RIOK2 inhibitors 5, 9 and 10 [22]. A construct containing RIOK2 residues 1–329 gave crystals when combined with inhibitor 9, and from one of these crystals a complete X-ray dataset was obtained to 2.35 Å resolution (table 1). From these data, it was possible to obtain an initial molecular replacement solution using the structure of *C. thermophilum* RIOK2, which after refinement and identification of additional RIOK2 molecules in the crystallographic asymmetric unit ultimately yielded a high-quality model of human RIOK2 with an $R_{free}$ of 24%. The final model contains 10 molecules of RIOK2 in the asymmetric unit.

### 2.2. Human RIOK2 structure

The overall structure of human RIOK2 closely resembles the structures of RIOK2 from the archaea *Archaeoglobus fulgidus* [17] and the fungus *Chaetomium thermophilum* [10] (figure 1). It has an ordered winged helix-turn-helix (wHTH) domain on the N-terminus followed by a kinase domain. The kinase domain is divided into two lobes, an N-terminal lobe (N-lobe) and a C-terminal lobe (C-lobe) which bind ATP between them. The sequence identity of the kinase domain of HsRIOK2 to those of CtRIOK2 and AfRIOK2 is 53% and 29%, respectively. Despite the closer relationship in sequence to CtRIOK2, HsRIOK2 most closely matches in overall three-dimensional structure the archaeal

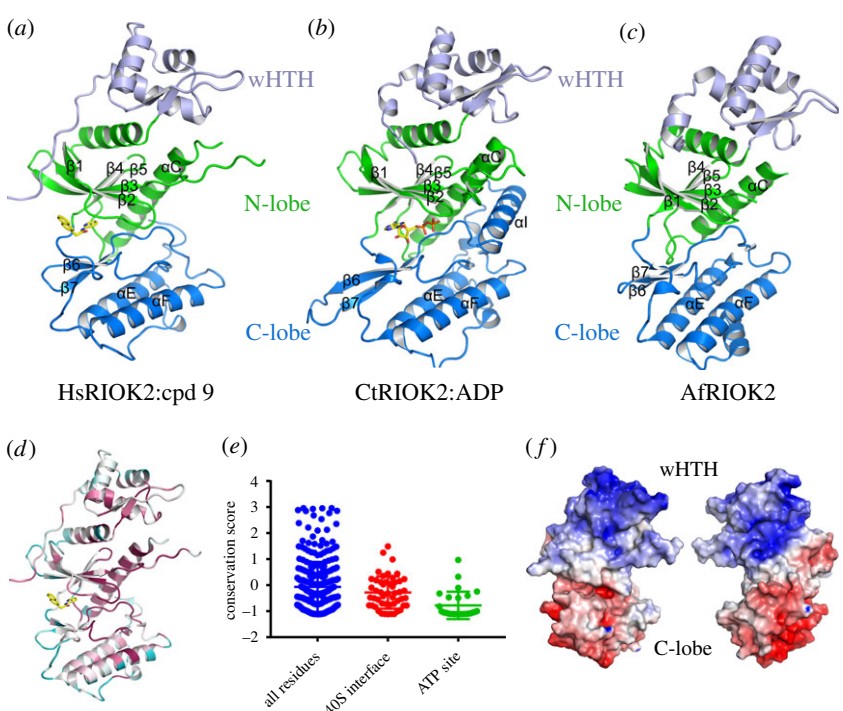

**Figure 1.** Human RIOK2 has an atypical kinase structure with significant differences from archaeal or fungal RIOK2, but is most closely related in overall structure to archaeal RIOK2 as exemplified by *Archaeoglobus fulgidus* RIOK2. (*a*) Structure of human RIOK2 with the N-terminal lobe of the kinase domain coloured yellow, the C-terminal lobe coloured blue and the wHTH domain coloured light blue. Compound 9 is shown with yellow carbon atoms bound in the ATP-binding site. (*b*) Structure of *Chaetomium thermophilum* RIOK2 [10] coloured as in (*a*) for comparison. The bound ADP molecule is shown with yellow carbon atoms. (*c*) Structure of *Archaeoglobus fulgidus* RIOK2 [17] coloured as in (*a*). (*d*) HsRIOK2 coloured by conservation of residues with homologous proteins identified by an HMMER search as implemented in ConSurf [24], from red as most conserved to blue as least conserved. (*e*) Conservation scores from (*d*) plotted for all residues, residues involved in interfaces with the pre-40S ribosome in PDB ID 6G51 (defined as within, and residues involved in binding or hydrolysing ATP. Horizontal bars represent the means and standard deviations. (*f*) Surface charge representation of hsRIOK2, calculated using a PARSE forcefield in PDB2PQR [25], plotted from $-6\,k_bT/e_c$ (red) to $+6\,k_bT/e_c$ (blue), showing the positively charged N-terminal wHTH domain for binding RNA.

AfRIOK2. Notable differences are the lack of an αI helix on the C-terminus of the kinase domain of HsRIOK2 and AfRIOK2 compared to the CtRIOK2 structure and the much shorter loop between β6 and β7 in HsRIOK2 compared to CtRIOK2. The N-terminus of the wHTH domain that extends over the N-terminal lobe of the kinase domain towards the ATP-binding site is also much more similar between the human and archaeal RIOK2 protein structures.

In the CtRIOK2 structure, the αI helix contains a C-terminal arginine, Arg342, that forms a salt bridge with Asp229 from the kinase catalytic HRD motif (HGD in both HsRIOK2 and CtRIOK2) and a hydrogen bond with Glu107 from the kinase P-loop. The region of ctRIOK2 containing αI (residues 324–341) is strongly predicted to be helical (using PSIPRED [26]; data not shown), but the equivalent region of HsRIOK2 (residues 306–323) is predicted to be coil and is disordered in the crystal structure. Taken together, it appears that the αI helix is not an evolutionarily conserved part of RIOK2 function.

The most relevant conformational difference seen in HsRIOK2 is in the ATP-binding P-loop formed by strands β1 and β2, which is folded around in the inhibitor in the human RIOK2 structure. This P-loop flexibility is a common feature of the protein kinases when binding different inhibitors, and since the ATP-binding functionality is conserved in RIOKs compared with the protein kinases it is expected that the P-loop of RIOKs might have similar flexibility to bind different types of inhibitor in place of ATP.

Analysis of the level of residue conservation (using an HMMER search as implemented in ConSurf [24]) shows that the most evolutionarily conserved regions of RIOK2 are buried in the interior of the protein (figure 1*d*). As it might be expected that regions of RIOK2 involved in binding to the pre-40S ribosome complex might be better conserved, we analysed the conservation score of subsets of RIOK2 residues (figure 1*e*). Defining residues involved in binding pre-40S ribosome as those RIOK2 residues within 5 Å distance of an adjacent protein or RNA component in structure 6G51 from the Protein Data Bank (PDB), a mean conservation score only marginally lower than the mean all residues score is obtained. For comparison, defining the ATP-binding site as all residues in contact with ATP or necessary for the catalytic mechanism (by comparison of HsRIOK2 to the structure of CtRIOK2 bound to ADP (PDB ID 4GYI)), it is clear that the residues of the ATP-binding site are significantly better conserved than the average.

Analysis of the surface charge distribution of RIOK2 reveals a positively charged N-terminal region, in particular the wHTH domain, and a negatively charged C-terminal lobe (figure 1*f*). The positively charged surface of the wHTH domain is in keeping with its predicted role as an RNA-binding domain, while the negatively charged C-lobe would be predicted to have minimal interactions with dsRNA. The cryo-EM structure of a yeast pre-40S particle matches this expectation, with the N-terminal region of Rio2 buried into the structure forming extensive interactions with RNA, while the C-lobe forms minimal interactions [27].

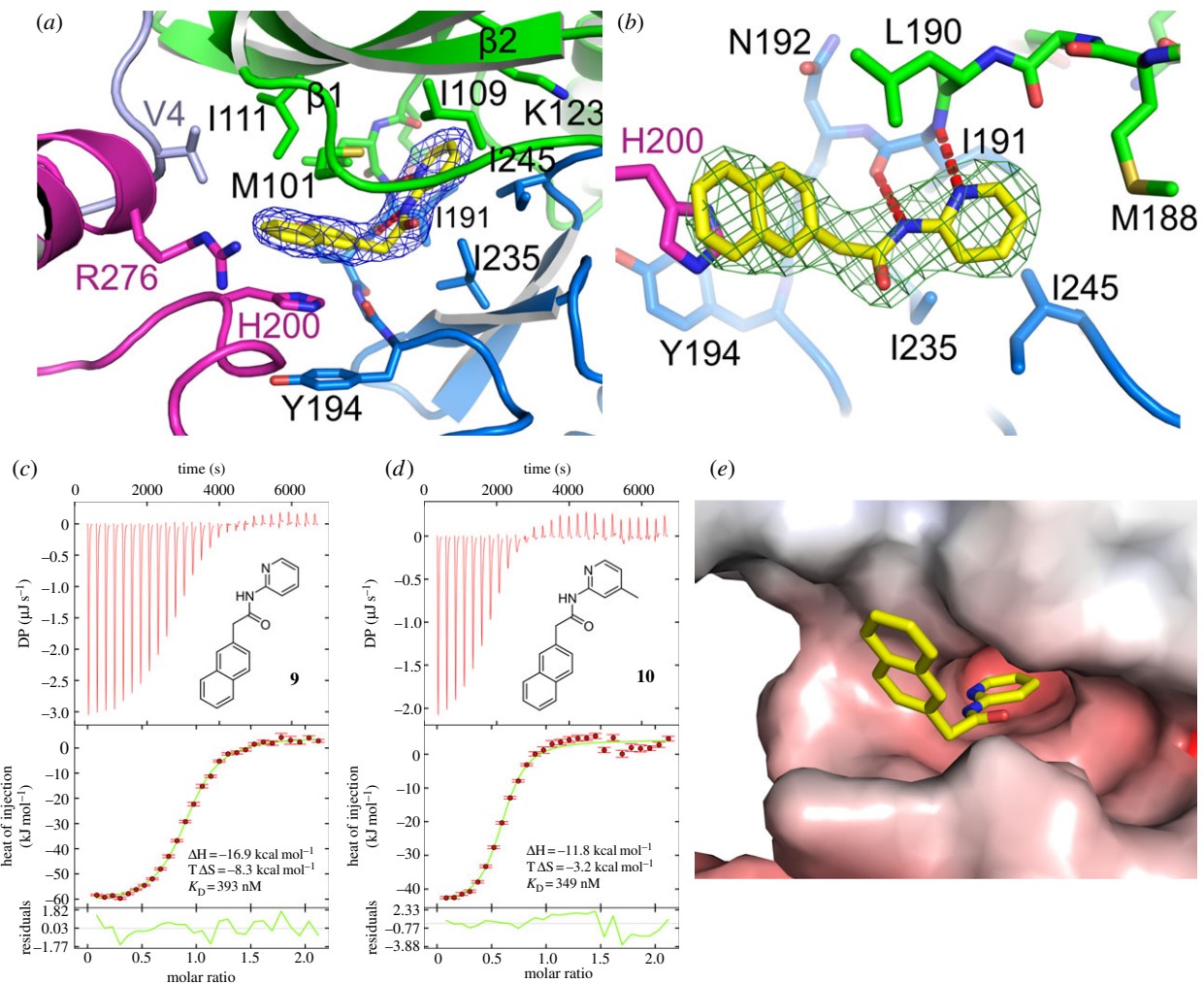

**Figure 2.** RIOK2 binds compound 9 in the ATP-binding site. RIOK2 is shown in green for the N-terminal lobe of the kinase domain, blue for the C-terminal lobe, light blue for the N-terminus, and adjacent RIOK2 molecules in the asymmetric unit are coloured magenta. (*a*) The inhibitor (shown in yellow) is bound in the ATP site, with two hydrogen bonds to the kinase hinge residue Ile191, and in nine out of ten molecules in the asymmetric unit the inhibitor also forms binding interactions with an adjacent RIOK2 molecule (shown in magenta) and the N-terminus of RIOK2 (shown in light blue). The $2F_o - F_c$ experimental electron density map is shown in blue mesh contoured at $1.0\sigma$ around the inhibitor, and the hydrogen bonds to RIOK2 at Ile191 are shown as red dashed lines. (*b*) The $F_o - F_c$ difference electron density map (calculated without 9 included in the model) is shown in green mesh contoured at $3.0\sigma$ around 9 to show the high confidence of placing 9 in the model. (*c,d*) Dissociation constants measured in solution by isothermal titration calorimetry show equivalent binding affinity to previously determined $IC_{50}$ values. The inhibitors bind with favourable enthalpy but unfavourable entropy. (*e*) Surface charge representation of the binding site for the inhibitor, showing the partial negative charge around the kinase hinge region that binds the pyridine moiety and the hydrophobic surface that binds the naphthalene moiety.

The loop between $\beta3$ and $\alpha$C is extended in RIOKs relative to many protein kinases and it is disordered in HsRIOK2. This disordered loop is hypothesized to be important for RNA interactions [10] and in the yeast pre-40S particle there is a cavity adjacent to the RNA which would accommodate this loop. In the structures of human pre-40S ribosome particles [20], the binding interfaces are less well-defined, including in one state an apparent lack of space for the disordered RIOK2 $\beta3$-$\alpha$C loop, and so while the general features of RIOK2 binding to the pre-40S complex are known, many details of the mechanism of action still remain to be resolved.

## 2.3. Inhibitor binding

The co-crystallized compound 9 is bound in the ATP-binding site of RIOK2 (figure 2*a*). The 2-aminopyridine moiety forms two hydrogen bonds with the hinge region of the kinase domain, to the backbone nitrogen and carbonyl oxygen of Ile191 (figures 2*a* and 3*b*). By contrast, ATP would bind with hydrogen bonds to the backbone carbonyl oxygen of Glu189 and the backbone nitrogen of Ile191, based on the structure of RIOK2:ADP from *Chaetomium thermophilum* [10]. The pyridine moiety is tightly enclosed by the hydrophobic side-chains of Met188, Ile109 and Ile245. There is only space for a small additional substituent at the 4-position of the pyridine ring (see below). The P-loop folds around the inhibitor such that Ile109 from strand $\beta2$ and Met101 from $\beta1$ can engage the top of the inhibitor.

The major reason for the selectivity of this inhibitor series for RIOK2 over RIOK1 and RIOK3 could be that Met101 is not conserved in RIOK1 and RIOK3 (which both have an isoleucine instead), and Ile111 is also not conserved (histidine). These two residues are critical for binding the naphthalene or biphenyl moieties of the inhibitors and their replacement with different amino acids presumably reduces significantly the binding affinity. At the pyridine end of the inhibitor, the gatekeeper residue Met188 is conserved in RIOK1 and RIOK3, as is Lys123. The replacement of Ile109 from RIOK2 with a valine in RIOK1 and RIOK3 may also be significant

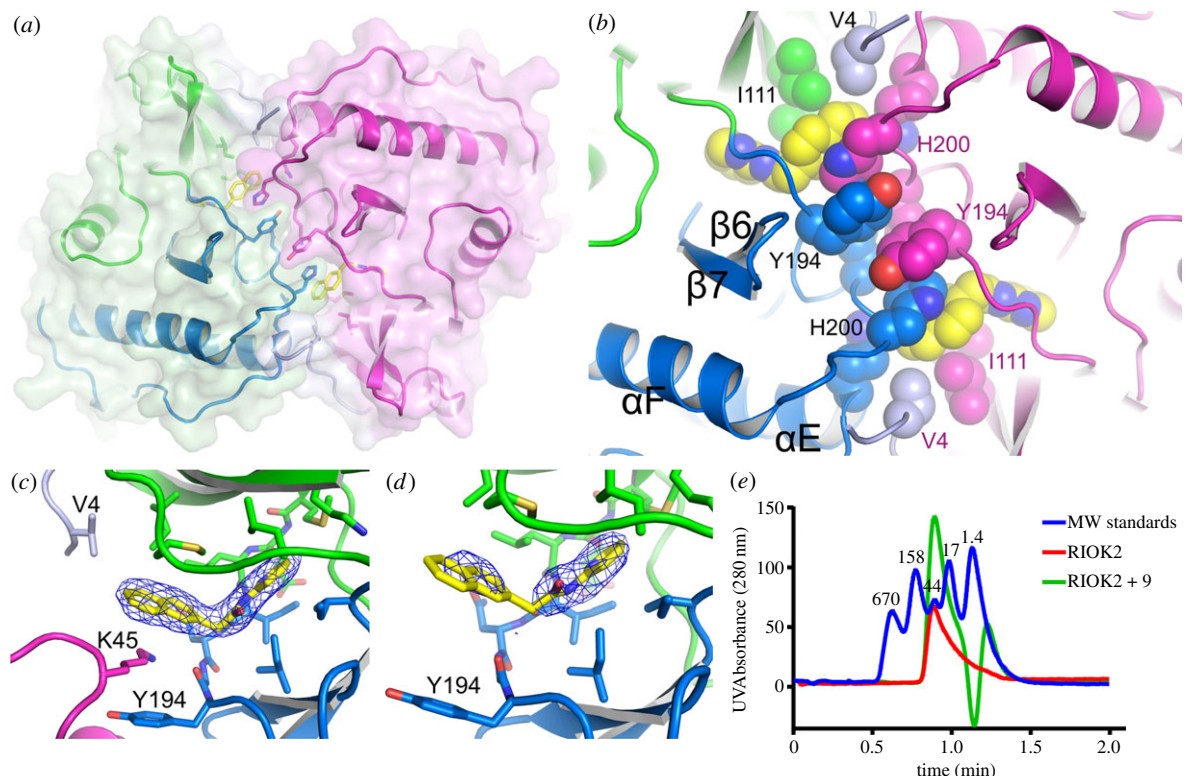

**Figure 3.** RIOK2 dimerizes around compound 9 in the crystal, but is monomeric in solution. RIOK2 is shown in green for the N-terminal lobe of the kinase domain, blue for the C-terminal lobe, light blue for the N-terminus and the adjacent RIOK2 molecule in the asymmetric unit is coloured magenta. (a) Eight of the ten RIOK2 molecules in the crystallographic asymmetric unit are involved in a twofold symmetrical dimeric interaction in which the naphthalene moiety of 9 is involved in an extended aromatic stacking interaction. (b) Detail of the core of the dimer interface showing the interactions of 9 with Ile111, His200 and Tyr194. (c) In the ninth molecule of RIOK2, His200 from an adjacent molecule in a twofold-symmetric dimer is replaced by Lys45, again from another RIOK2 molecule, but not related by twofold symmetry. (d) In the tenth molecule of RIOK2, there is no residue equivalent to His200 or Lys45 binding to 9, and the electron density for the naphthalene moiety is poorly defined. (e) Analytical size-exclusion chromatography analysis of RIOK2 in the presence or the absence of 9. RIOK2 elutes with the same retention time as the 44 kDa standard protein (ovalbumin) irrespective of the presence of the inhibitor, indicating that RIOK2 is monomeric. The absorbance was measured at 280 nm wavelength and data are baseline-corrected. Molecular weights of the standard calibration proteins are shown in kDa above the trace for the standard proteins.

as the additional methyl group of Ile109 is involved directly in binding the pyridine moiety.

To confirm the potency of the inhibitors against RIOK2 *in vitro* and to characterize the thermodynamics of binding, we measured dissociation constants ($K_D$) using isothermal titration calorimetry (ITC). Compound 9 had a $K_D$ of 393 nM and compound 10 had a $K_D$ of 349 nM (figure 2c, d). Interestingly, the binding was strongly enthalpic, with very favourable $\Delta H$ values of $-16.9$ kcal mol$^{-1}$ and $-11.8$ kcal mol$^{-1}$ for compounds 9 and 10, respectively, while having unfavourable entropy of binding. Part of the enthalpy of binding is derived from the two hydrogen bonds to kinase hinge region (2.9 Å and 3.0 Å distance). This, combined with the poorer electron density for the naphthalene moiety in the absence of an adjacent RIOK2 molecule in the asymmetric unit (figure 3d), suggests that the 2-aminopyridine core of the inhibitors provides a significant part of the binding affinity.

We then attempted to measure the affinity of compound 9 for RIOK2 in intact live cells using the NanoBRET method (Promega). HEK293 cells were transfected with a fusion of the Nanoluc luciferase and the full-length RIOK2 gene. Using a fluorescent tracer molecule (Promega), we were able to generate a BRET signal from intact cells in the presence of the Nanoluc substrate. However, compound 9 did not displace the tracer molecule from RIOK2 at the concentrations tested (up to 50 μM; data not shown). Compound 9

is predicted to be highly permeable in MDCK cells by Lilly internal methods, with 95% predicted protein binding, and therefore it is likely that competition with cellular ATP requires a more potent inhibitor to be effective in cells.

Examination of the surface charge distribution of RIOK2 around the inhibitor reveals that the hinge region of RIOK2 that binds the pyridine moiety is partially negatively charged as expected, while there is a large hydrophobic surface at the entrance to the ATP-binding site, against which the naphthalene moiety binds (figure 2e).

## 2.4. RIOK2 is monomeric in solution including in the presence of the inhibitor

Eight of the ten RIOK2 molecules in the crystallographic asymmetric unit, eight molecules were seen as four RIOK2 dimers, each forming the same dimeric arrangement with a twofold rotational symmetry (figure 3a). This dimeric arrangement has a large interaction interface that involves the binding of compound 9 in an extended aromatic stacking interaction that includes His200 and Tyr194 for six stacked aromatic rings (figure 3b). Of the remaining two RIOK2 molecules in the asymmetric unit, one showed a different interaction with a neighbouring molecule in the crystal lattice that uses the side-chain of Lys45 in place of His200 (figure 3c). The

royalsocietypublishing.org/journal/rsob   Open Biol. 9: 190037

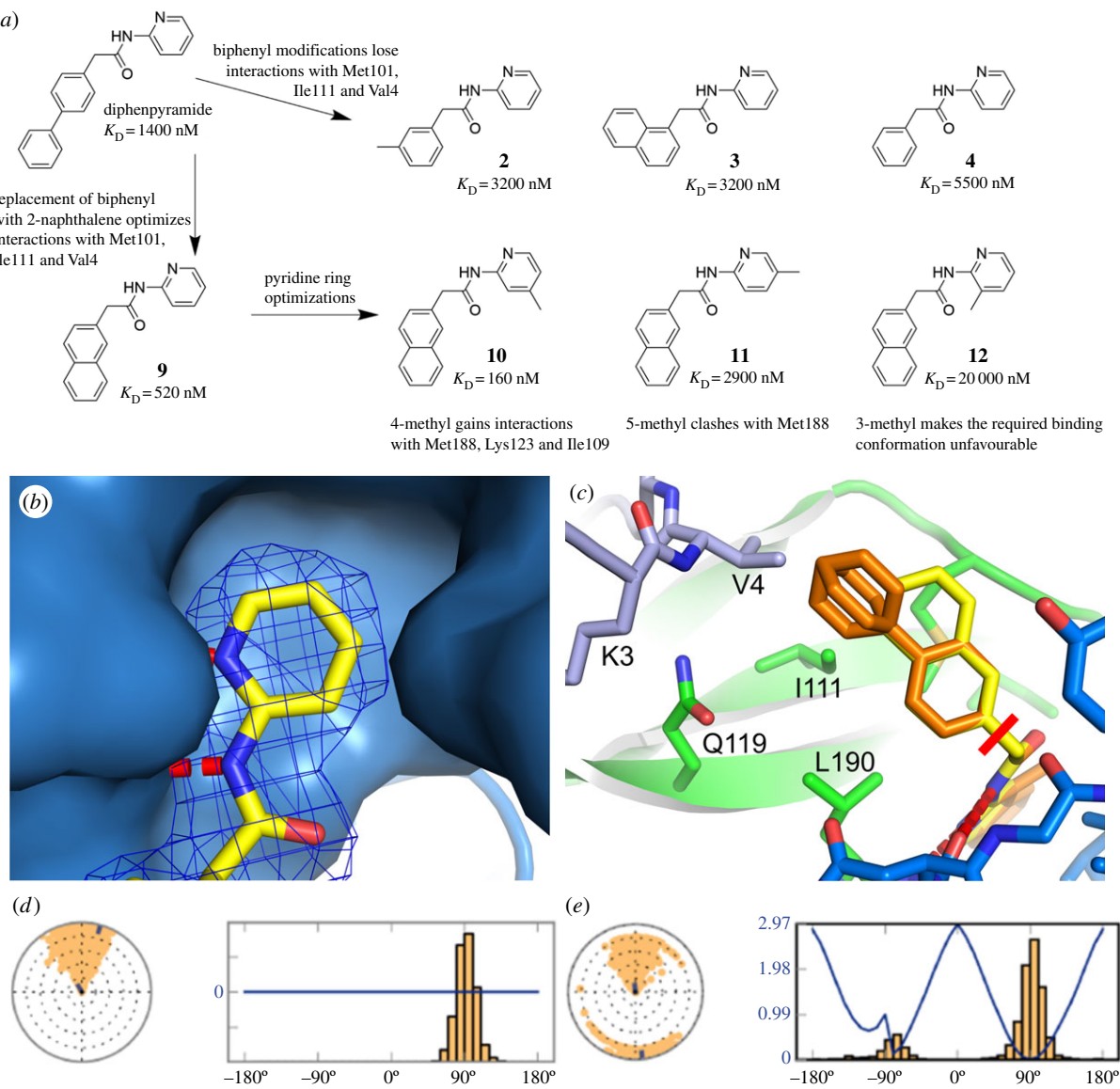

**Figure 4.** Structure–activity relationship of the inhibitor series. Compound numbering is the same as that of Varin *et al*. [22]. (*a*) Replacement of the biphenyl moiety with a 2-substituted naphthalene optimizes interactions with residues at the entrance to the ATP-binding site, while the 4-methyl substituent on the pyridine enables favourable hydrophobic interactions with Met101, Ile111 and Val4. (*b*) The pyridine ring of 9 is bound in a tight pocket, with space only for the extra 4-methyl substituent of compound 10. A 5-methyl substituent as in compound 11 can only fit with the inhibitor or the protein adopting a slightly different conformation. Compound 9 is shown with yellow carbon atoms and a molecular surface around RIOK2 is shown in blue. (*c*) Binding of compounds with the biphenyl moiety would require movement of Val4 and Ile111. (*d,e*) A 200 ns molecular dynamic simulation shows that in the absence of His200 (figure 3) from a dimeric interaction, there is a much greater distribution of rotation angles for the naphthalene moiety (*e*), compared to with His200 (*d*). The radial plots show the time evolution of the torsion angle (marked on the structure with a red line in (*c*)) starting from time zero in the centre, and the bar plots summarize the radial plots by showing the probability density of the torsion angle.

remaining RIOK2 molecule was not bound next to another RIOK2 molecule in the crystal and did not have an alternative to the binding of His200 or Lys45 between the naphthalene of compound 9 and Tyr194. In this molecule, the electron density was weak for the naphthalene of compound 9 (figure 3*d*), and also for other residues at the interaction interface seen in the other RIOK2 molecules. As compound 9 appears at the interface between RIOK2 molecules in the crystal, it may have been essential for the successful crystallization.

As a residue from a second RIOK2 molecule appeared important for binding compound 9 (figure 4*b,c*), and in the absence of such a residue compound 9 was less well ordered in the structure, the data suggested that dimerization might be important for binding this inhibitor series. Therefore, we wanted to understand if RIOK2 was indeed a dimer in

solution, or if the binding of the inhibitor used in the crystal structure would promote dimerization. We performed analytical size-exclusion chromatography, measuring the molecular size of RIOK2 (residues 1–329, as in the crystal) in the presence and absence of compound 9, compared to a set of standard size proteins. RIOK2 alone matched exactly the expected size for a monomeric protein, eluting at the same retention time as the 44 kDa standard protein, and the presence of compound 9 did not alter this retention time (figure 3*e*). From these data, we conclude that the observed dimer formed by eight of the ten RIOK2 molecules in the asymmetric unit is likely formed only at the high RIOK2 concentrations that occur during crystallization, possibly promoted by inhibitor binding, but at concentrations achievable in solution RIOK2 remains monomeric, even in the presence of the inhibitor.

royalsocietypublishing.org/journal/rsob    Open Biol. 9: 190037

# 3. Discussion

The structure of RIOK2 reveals the binding mode of the first-ever RIOK-specific inhibitor and highlights opportunities for the design of improved molecules. The 2-aminopyridine moiety binds the kinase hinge region as could have been predicted, but the need for a biphenyl or naphthalene moiety in particular is only clear now that the structure reveals an extended hydrophobic surface on the protein at the entrance to the ATP-binding site (figure 2e). The previously determined dissociation constants for this inhibitor series [22] using the DiscoverX KdELECT measurement method revealed that on the pyridine ring only a 4-methyl substituent gave improved binding (about threefold reduced $K_D$), while of the biphenyl replacements that were tested, only a naphthalene joined at its 2-position gave improved binding (also about threefold reduced $K_D$) (figure 4a). The pyridine ring sits in a tight pocket which has space only for the addition of a methyl group on position 4 of the ring without significantly altering the observed binding conformation (figure 4b); a small adjustment, mainly of residue Ile109, would be sufficient to allow the additional methyl. A 5-methyl substitution on the pyridine reduces binding affinity by 5–9-fold (comparison of pairs of compounds 1–6 and 9–11), while a 3-methyl substitution is not tolerated (comparison of pairs of compounds 1–7 and 9–12). Replacement of the naphthalene by the biphenyl moiety reduces potency by around threefold (comparisons of pairs of compounds 9–1, 10–5 and 11–6). This can be explained by additional hydrophobic interactions for the naphthalene and/or by the close proximity of Val4 with the biphenyl. The Val4 in the conformation observed with compound 9 (naphthalene) clashes with the biphenyl ring (figure 4c).

The dimerization interface observed in this crystal form, and especially the presence of a side-chain from the adjacent RIOK2 molecule binding between the naphthalene moiety and Tyr194 (figure 3b, 4c), suggests that in solution, in the absence this interaction, the naphthalene moiety could be rotated 180° to interact better with Met101. Molecular dynamic simulations show a differential torsional profile for the naphthyl in the dimer where it oscillates around the position seen in the X-ray structure compared to the monomer in which there is some 180° flipping of the naphthyl (figure 4d; electronic supplementary material, figures S1 and S2). In this case, there would be many possibilities for modifying the inhibitors to optimize interactions with both Met101 and Tyr194.

The various cryo-EM structures of pre-40S ribosomes show RIOK2 (named Rio2 in the case of yeast structures) bound to the other components of the pre-40S complex in a variety of conformations [18–20]. For example in the yeast pre-40S particle, Rio2 is likely unable to bind ATP due to a rotation of its N- and C-terminal lobes [27]. Given this heterogeneity, it is difficult to predict the effects of chemical inhibition on the different stages of ribosome assembly. Various questions arise, such as how significantly would chemical inhibition of RIOK2 impede ribosome assembly given that at any inhibitor concentration there would always be at least a small proportion of free RIOK2. What proportion of free RIOK2 to pre-40S-bound RIOK2 is there in the cell, and how much can this be perturbed before ribosome assembly is impaired? Would the RIOK2 inhibitors presented here allow binding of RIOK2

in the pre-40S complex? The development of more potent RIOK2 inhibitors capable of binding RIOK2 in cells may be able to answer some of these questions as well as help to identify other functions of RIOK2 apart from ribosome assembly. The structure of RIOK2 bound to compound 9 presented here provides a starting point for generating future inhibitor series.

# 4. Methods

## 4.1. Cloning, protein expression and purification

DNA for residues 1–329 of human RIOK2 isoform 1 (NP_060813.2) was PCR amplified from DNA in the Mammalian Gene Collection (IMAGE consortium clone ID 3449997) and inserted into the vector pNIC-ZB. The resulting construct expresses an N-terminal hexahistidine tag, a Zbasic tag, a TEV (tobacco etch virus) protease tag cleavage site followed by the RIOK2 kinase domain. The construct was verified by DNA sequencing.

The construct was transformed into BL21(DE3) cells that contained the pRARE2 plasmid from commercial Rosetta2 cells. The resulting colonies were used to inoculate 50 ml of LB media containing 50 µg ml$^{-1}$ kanamycin and 34 µg ml$^{-1}$ chloramphenicol, which was left shaking at 37°C overnight. This culture was used to inoculate 1 l volumes of LB media containing 33 µg ml$^{-1}$ kanamycin at a ratio of 10 ml culture to 1 l fresh media. The cultures were grown at 37°C with shaking until an OD600 of 0.8–0.9 was reached. The temperature was reduced to 18°C, and isopropyl β-D-1-thiogalactopyranoside was added to a final concentration of 0.5 mM and the cultures were left overnight. Cells were harvested by centrifugation, re-suspended in Lysis Buffer (50 mM Hepes pH 7.5, 200 mM NaCl, 20 mM imidazole, 5% glycerol, 0.5 mM tris(2-carboxyethyl)phosphine (TCEP) and a protease inhibitor cocktail (Sigma)).

The re-suspended cells were lysed by sonication, polyethylenimine was added to a final concentration of 0.15%, and the insoluble debris was removed by centrifugation. The supernatant was passed through a column of 6 ml Ni-Sepharose resin (GE Healthcare) at room temperature. The resin was washed with Binding Buffer (50 mM Hepes pH 7.5, 500 mM NaCl, 5 mM imidazole, 5% glycerol, 20 mM arginine, 20 mM glutamate, 0.5 mM TCEP) containing increasing amounts of imidazole before elution with Binding Buffer containing 250 mM imidazole. TEV protease was added to the eluate, which was dialysed into GF Buffer (20 mM Hepes pH 7.5, 500 mM NaCl, 5% glycerol, 20 mM arginine, 20 mM glutamate, 0.5 mM TCEP) overnight at 4°C. The protein was further purified by passing through a column of Ni-Sepharose. The column was washed with GF Buffer containing increasing amounts of imidazole. The desired RIOK2 protein was eluted in a fraction containing 60 mM imidazole. The fractions containing RIOK2 were concentrated to 5 ml (volume) and injected on a S75 16/60 gel filtration column (GE Healthcare) pre-equilibrated into GF Buffer. Protein identity was confirmed by electrospray ionization mass spectrometry (expected 38 205.9 Da, observed 38 207.8 Da). The purified protein was concentrated to 14 mg ml$^{-1}$ (measured by UV absorbance, using the calculated molecular weight and estimated extinction coefficient, using a NanoDrop spectrophotometer (Thermo Scientific)) and used for crystallization.

royalsocietypublishing.org/journal/rsob   Open Biol. 9: 190037

## 4.2. Crystallization and structure determination

For crystallization trials, the inhibitors were added to the concentrated protein to a final concentration of 1 mM. Resultant insoluble material was removed by centrifugation. Crystals were obtained using the sitting-drop vapour diffusion method at 20°C. Crystals grew from a mixture of 100 nl protein and 50 nl of a well solution containing 20% PEG3350, 10% ethylene glycol, 0.1 M bis-tris-propane pH 6.5 and 0.2 M sodium/potassium tartrate. Crystals were equilibrated into the reservoir solution plus 25% ethylene glycol before freezing in liquid nitrogen. Data were collected at 100 K at the Diamond Synchrotron beamline I02. Data collection statistics can be found in table 1.

The diffraction data were processed using MOSFLM [28] and AIMLESS [29]. The RIOK2 structure was solved by molecular replacement using PHASER [30] and a truncated version of the structure of *Chaetomium thermophilum* RIO2 kinase as a search model (PDB ID 4GYI [10]). Initially, seven molecules were identified in the asymmetric unit. After several cycles of model building using Coot [31] and refinement with REFMAC5 [32], the improved model was used for a second cycle of molecular replacement searching, which identified 10 molecules in the asymmetric unit. Further cycles of model building and refinement resulted in the final model. MOLPROBITY [33] was used for model validation and analysis.

## 4.3. Analytical size-exclusion chromatography

Samples were injected onto an S200 5/150 size-exclusion chromatography column (cross-linked agarose/dextran matrix, $5 \times 150$ mm diameter×height, 3 ml bed volume, GE Healthcare). For measurement of the retention time of RIOK2 alone, the column was pre-equilibrated in GF Buffer and the RIOK2 protein sample was in GF Buffer before injection. For measurement of the retention time of RIOK2 + compound 9, the column was pre-equilibrated in GF Buffer + 20 µM compound 9 and the RIOK2 protein sample was in GF Buffer + 20 µM compound 9 before injection. The molecular weight standards were as supplied (BioRad). Absorbance was measured at 280 nm.

## 4.4. Isothermal titration calorimetry

RIOK2 protein at 10.5 mg ml$^{-1}$ was dialysed overnight into ITC Buffer (20 mM HEPES pH 7.5, 500 mM NaCl, 5% glycerol, 20 mM arginine, 20 mM glutamate, 0.5 mM TCEP) at 4°C. For measurements, a VP-ITC instrument (GE Healthcare) was used with a cell temperature of 20°C. For each measurement, the cell contained inhibitor at 16 µM dissolved in the dialysis buffer, with final DMSO concentration of 0.8%, and the syringe contained RIOK2 protein at 160 µM with DMSO added to 0.8%. $1 \times 2$ µl and $27 \times 10$ µl injections were made, with 240 s spacing. The data were analysed using NITPIC [34], SEDPHAT [35] and GUSSI [36]. The measurement with compound 9 had a fraction of incompetent protein = 0.065 and a fraction of incompetent ligand 0.157, and the measurement with compound 10 had a fraction of incompetent protein = 0.000 and a fraction of incompetent ligand = 0.418.

## 4.5. NanoBRET assay

HEK293 cells were maintained at 70% confluency. The DNA solution for transfection consisted of 4.5 µg ml$^{-1}$ Transfection Carrier DNA (Promega) and 0.5 µg ml$^{-1}$ of Nanoluc-RIOK2 fusion DNA (Promega) in 0.5 ml of Opti-MEM media without phenol red. This solution was mixed with 15 µl of FuGENE transfection reagent (Promega) and then used to transfect 10 ml of cells at $2 \times 10^5$ cells ml$^{-1}$ for 20 h. These cells were used to perform the NanoBRET target engagement assay according to the manufacturer's instructions (Promega). A 14-point twofold serial dilution of compound 9 starting at 50 µM final assay concentration was used. Two concentrations of Tracer 5 (Promega) were tested, 1.0 µM and 2.0 µM.

Measurements were made in a PheraStar FS plate reader (BMG Labtech) at 450 and 610 nm 10 min after adding the substrate. The readings at 610 nm were divided by the readings at 450 nm to calculate the BRET ratio. The BRET ratio for control wells without tracer was subtracted from all other BRET ratios, and the results were multiplied by 1000 to give the final values (NanoBRET units).

Data accessibility. The coordinates and structure factors for the crystal structure reported in this article have been deposited in the PDB with accession code 6HK6.

Authors' contributions. T.V., M.V. and J.M.E. conceived the study. J.W. carried out the protein-based laboratory work and participated in data analysis. J.M.E. carried out crystal structure determination. T.V. performed MD simulations. J.M.E. wrote the manuscript with assistance from all authors. All authors gave final approval for publication.

Competing interests. T.V. and M.V. are employees of Eli Lilly and Company.

Funding. J.W. and J.M.E. are supported by the SGC, is a registered charity (number 1097737) that receives funds from AbbVie, Bayer Pharma AG, Boehringer Ingelheim, Canada Foundation for Innovation, Eshelman Institute for Innovation, Genome Canada, Innovative Medicines Initiative (EU/EFPIA) (ULTRA-DD grant no. 115766), Janssen, Merck KGaA Darmstadt Germany, MSD, Novartis Pharma AG, Ontario Ministry of Economic Development and Innovation, Pfizer, São Paulo Research Foundation-FAPESP, Takeda and Wellcome (106169/ZZ14/Z). T.V. and M.V. are supported by Eli Lilly and Company.

Acknowledgements. We thank Dr Prashant Desai for discussions on the permeability and properties of compound 9.

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
