## [Reviewer comments · Open Biology]

Review History

RSOB-19-0037.R0 (Original submission)

Review form: Reviewer 1

Recommendation

Accept with minor revision (please list in comments)

Are each of the following suitable for general readers?

- a) **Title**
Yes
- b) **Summary**
Yes
- c) **Introduction**
Yes

Is the length of the paper justified?

Yes

Should the paper be seen by a specialist statistical reviewer?

No

Is it clear how to make all supporting data available?

Yes

Is the supplementary material necessary; and if so is it adequate and clear?

Yes

Do you have any ethical concerns with this paper?

No

Comments to the Author

The paper reports the first crystal structure of human RIOK2. In the structure, the enzyme is in complex with a representative of a series of RIOK2-binding ligands that were identified previously by some of the authors (Varin et al. BBA 1854:1630-1636).

The structure is of excellent quality and, being the first of human RIOK2, constitutes a significant advance. The analysis of the structure reported in the manuscript is well done and worthy of publication.

The ligands are referred to in the title and throughout the paper as 'inhibitors'. Perhaps the term 'inhibitor' is not appropriate since no data is presented that shows these compounds inhibit an aspect of the enzyme's activity.

Since the naphthalene moiety of ligand 9 is interposed at non-crystallographic intermolecular interfaces at several places in the asymmetric unit (and no apo- or nucleotide-bound RIOK2 structure is presented) it seems likely that the ligand has in fact been crystallogenic. If this is the case, the authors should comment upon this.

The ligand seems to be bound primarily via a pyridine group and the hinge-binding motif while the distal portion of the ligand lies at the mouth of the nucleotide-binding pocket and is not well-ordered in the crystal structure except when it is stabilized by an adjacent molecule. The authors claim that this portion of the ligand (a naphthalene group in ligand 9) forms 'extensive hydrophobic interactions with residues at the entrance to the ATP-binding site'. However, there is doubt about the extent to which the position and interactions of the naphthalene group in the structure reflects those of RIOK2-ligand complexes in solution and MD simulations suggest some mobility. This group is apparently responsible for much of the selectivity of the ligand, so the lack of clarity about it is unfortunate, however the uncertainty is investigated to a reasonable degree and is described well in the manuscript.

The manuscript might be improved by generating the M101I, I111H and I109V mutants of RIOK2, and perhaps the triple mutant, to mimic RIOK1/3 and measuring the affinities of these mutants for ligand 9 in order to investigate experimentally whether, as the authors propose, these residues are molecular determinants of selectivity for RIOK2 over RIOK1 and 3.

Review form: Reviewer 2

Recommendation

Accept with minor revision (please list in comments)

Are each of the following suitable for general readers?

- a) **Title**
Yes
- b) **Summary**
Yes
- c) **Introduction**
Yes

Is the length of the paper justified?

Yes

Should the paper be seen by a specialist statistical reviewer?

No

Is it clear how to make all supporting data available?

Yes

Is the supplementary material necessary; and if so is it adequate and clear?

Yes

Do you have any ethical concerns with this paper?

No

Comments to the Author

In the paper 'Crystal structure of human RIOK2 bound to a specific inhibitor' submitted by Jing Wang et al. The authors report the first crystal structure of an inhibitor to the atypical kinase RIOK2. Here the authors implement protein purification, crystallisation, X-ray structural determination, ITC, fluorescence based cellular assay, and molecular dynamics. The authors present a suitable body of work, that is well designed and executed. The MS is also well written and a nice light read. The study is of interest to the kinase/small molecule community

Introduction

Line 77/78 - use of 'Rio2' I assume this is how RIOK2 is referred to in (18)?

Results

Line 95 - remove the word 'protein's'

Line 153 - 'Rio2' usage again

Fig.2 - In C & D could you rotate the chem draw pyridine to mimic its binding mode, i.e both N on same side (I know it is small thing just looks neater) as you have drawn it in Fig.4

Discussion

Line 239 - 'The previously determined dissociation constants for this inhibitor series.....' I am mildly confused with this and the origins of the SAR (Varin et al) when I read the paper in

absence of referring to the Varin et al work. How I review this is that the SAR are both against RIOK2, the reported Kd in this paper for compounds 9 and 10 are determined via ITC. But the Varin et al SAR is not clear the method used to obtain Kd's in this paper. Need to reference the KdELECT DiscoverX displacement method here, otherwise, hard to compare the 393 and 349nM reported from ITC for compounds 9 and 10 against Fig.4 with 520 and 160nM for 9 and 10, respectively. I have no problem with the measurements and the authors interpretations, I would just like a little clarity of the origins.

Decision letter (RSOB-19-0037.R0)

12-Mar-2019

Dear Dr Elkins,

We are pleased to inform you that your manuscript RSOB-19-0037 entitled "Crystal structure of human RIOK2 bound to a specific inhibitor" has been accepted by the Editor for publication in Open Biology. The reviewer(s) have recommended publication, but also suggest some minor revisions to your manuscript. Therefore, we invite you to respond to the reviewer(s)' comments and revise your manuscript.

Please submit the revised version of your manuscript within 7 days. If you do not think you will be able to meet this date please let us know immediately and we can extend this deadline for you.

- 1) A text file of the manuscript (doc, txt, rtf or tex), including the references, tables (including captions) and figure captions. Please remove any tracked changes from the text before submission. PDF files are not an accepted format for the "Main Document".
- 2) A separate electronic file of each figure (tiff, EPS or print-quality PDF preferred). The format should be produced directly from original creation package, or original software format. Please note that PowerPoint files are not accepted.
- 3) Electronic supplementary material: this should be contained in a separate file from the main

text and meet our ESM criteria (see <http://royalsocietypublishing.org/instructions-authors#question5>). All supplementary materials accompanying an accepted article will be treated as in their final form. They will be published alongside the paper on the journal website and posted on the online figshare repository. Files on figshare will be made available approximately one week before the accompanying article so that the supplementary material can be attributed a unique DOI.

Online supplementary material will also carry the title and description provided during submission, so please ensure these are accurate and informative. Note that the Royal Society will not edit or typeset supplementary material and it will be hosted as provided. Please ensure that the supplementary material includes the paper details (authors, title, journal name, article DOI). Your article DOI will be 10.1098/rsob.2016[last 4 digits of e.g. 10.1098/rsob.20160049].

4) A media summary: a short non-technical summary (up to 100 words) of the key findings/importance of your manuscript. Please try to write in simple English, avoid jargon, explain the importance of the topic, outline the main implications and describe why this topic is newsworthy.

Images

Data-Sharing

It is a condition of publication that data supporting your paper are made available. Data should be made available either in the electronic supplementary material or through an appropriate repository. Details of how to access data should be included in your paper. Please see <http://royalsocietypublishing.org/site/authors/policy.xhtml#question6> for more details.

Data accessibility section

Sincerely,

The Open Biology Team
<mailto:openbiology@royalsociety.org>

Reviewer(s)' Comments to Author:

Referee: 1

Comments to the Author(s)

The paper reports the first crystal structure of human RIOK2. In the structure, the enzyme is in complex with a representative of a series of RIOK2-binding ligands that were identified previously by some of the authors (Varin et al. BBA 1854:1630-1636).

The structure is of excellent quality and, being the first of human RIOK2, constitutes a significant advance. The analysis of the structure reported in the manuscript is well done and worthy of publication.

The ligands are referred to in the title and throughout the paper as 'inhibitors'. Perhaps the term 'inhibitor' is not appropriate since no data is presented that shows these compounds inhibit an aspect of the enzyme's activity.

Since the naphthalene moiety of ligand 9 is interposed at non-crystallographic intermolecular interfaces at several places in the asymmetric unit (and no apo- or nucleotide-bound RIOK2 structure is presented) it seems likely that the ligand has in fact been crystallogenic. If this is the case, the authors should comment upon this.

The ligand seems to be bound primarily via a pyridine group and the hinge-binding motif while the distal portion of the ligand lies at the mouth of the nucleotide-binding pocket and is not well-ordered in the crystal structure except when it is stabilized by an adjacent molecule. The authors claim that this portion of the ligand (a naphthalene group in ligand 9) forms 'extensive hydrophobic interactions with residues at the entrance to the ATP-binding site'. However, there is doubt about the extent to which the position and interactions of the naphthalene group in the structure reflects those of RIOK2-ligand complexes in solution and MD simulations suggest some mobility. This group is apparently responsible for much of the selectivity of the ligand, so the lack of clarity about it is unfortunate, however the uncertainty is investigated to a reasonable degree and is described well in the manuscript.

The manuscript might be improved by generating the M101I, I111H and I109V mutants of RIOK2, and perhaps the triple mutant, to mimic RIOK1/3 and measuring the affinities of these mutants for ligand 9 in order to investigate experimentally whether, as the authors propose, these residues are molecular determinants of selectivity for RIOK2 over RIOK1 and 3.

Referee: 2

Comments to the Author(s)

In the paper 'Crystal structure of human RIOK2 bound to a specific inhibitor' submitted by Jing Wang et al. The authors report the first crystal structure of an inhibitor to the atypical kinase RIOK2. Here the authors implement protein purification, crystallisation, X-ray structural determination, ITC, fluorescence based cellular assay, and molecular dynamics. The authors present a suitable body of work, that is well designed and executed. The MS is also well written and a nice light read. The study is of interest to the kinase/small molecule community

Introduction

Line 77/78 - use of 'Rio2' I assume this is how RIOK2 is referred to in (18)?

Results

Line 95 – remove the word ‘protein’s’

Line 153 – ‘Rio2’ usage again

Fig.2 – In C & D could you rotate the chem draw pyridine to mimic its binding mode, i.e both N on same side (I know it is small thing just looks neater) as you have drawn it in Fig.4

Discussion

Line 239 – ‘The previously determined dissociation constants for this inhibitor series.....’ I am mildly confused with this and the origins of the SAR (Varin et al) when I read the paper in absence of referring to the Varin et al work. How I review this is that the SAR are both against RIOK2, the reported Kd in this paper for compounds 9 and 10 are determined via ITC. But the Varin et al SAR is not clear the method used to obtain Kd’s in this paper. Need to reference the KdELECT DiscoverX displacement method here, otherwise, hard to compare the 393 and 349nM reported from ITC for compounds 9 and 10 against Fig.4 with 520 and 160nM for 9 and 10, respectively. I have no problem with the measurements and the authors interpretations, I would just like a little clarity of the origins.

Author's Response to Decision Letter for (RSOB-19-0037.R0)

See Appendix A.

Decision letter (RSOB-19-0037.R1)

20-Mar-2019

Dear Dr Elkins

We are pleased to inform you that your manuscript entitled "Crystal structure of human RIOK2 bound to a specific inhibitor" has been accepted by the Editor for publication in Open Biology.

Article processing charge

Please note that the article processing charge is immediately payable. A separate email will be sent out shortly to confirm the charge due. The preferred payment method is by credit card; however, other payment options are available.

Sincerely,

The Open Biology Team
mailto: openbiology@royalsociety.org

Appendix A

Referee: 1

Comments to the Author(s)

The paper reports the first crystal structure of human RIOK2. In the structure, the enzyme is in complex with a representative of a series of RIOK2-binding ligands that were identified previously by some of the authors (Varin et al. BBA 1854:1630-1636).

The structure is of excellent quality and, being the first of human RIOK2, constitutes a significant advance. The analysis of the structure reported in the manuscript is well done and worthy of publication.

We thank the reviewer for the positive evaluation and address the individual remarks below.

The ligands are referred to in the title and throughout the paper as 'inhibitors'. Perhaps the term 'inhibitor' is not appropriate since no data is presented that shows these compounds inhibit an aspect of the enzyme's activity.

It is true that our manuscript and the previous publication concerning this inhibitor series show binding assays rather than an assay to measure ATP consumption or ADP production, however since the molecules bind in the ATP binding site we think it is a reasonable conclusion that they inhibit the enzymatic activity of RIOK2.

Since the naphthalene moiety of ligand 9 is interposed at non-crystallographic intermolecular interfaces at several places in the asymmetric unit (and no apo- or nucleotide-bound RIOK2 structure is presented) it seems likely that the ligand has in fact been crystallogenic. If this is the case, the authors should comment upon this.

We have added a sentence "As compound 9 appears at the interface between RIOK2 molecules in the crystal it may have been essential for the successful crystallisation." To the end of the first paragraph of the results section "RIOK2 is monomeric in solution including in the presence of the inhibitor" on page 11.

The ligand seems to be bound primarily via a pyridine group and the hinge-binding motif while the distal portion of the ligand lies at the mouth of the nucleotide-binding pocket and is not well-ordered in the crystal structure except when it is stabilized by an adjacent molecule. The authors claim that this portion of the ligand (a naphthalene group in ligand 9) forms 'extensive hydrophobic interactions with residues at the entrance to the ATP-binding site'. However, there is doubt about the extent to which the position and interactions of the naphthalene group in the structure reflects those of RIOK2-ligand complexes in solution and MD simulations suggest some mobility. This group is apparently responsible for much of the

selectivity of the ligand, so the lack of clarity about it is unfortunate, however the uncertainty is investigated to a reasonable degree and is described well in the manuscript.

We agree that it is a difficult question to resolve: how exactly does the naphthalene contribute to the compound binding affinity in terms of binding energetics.

The manuscript might be improved by generating the M101I, I111H and I109V mutants of RIOK2, and perhaps the triple mutant, to mimic RIOK1/3 and measuring the affinities of these mutants for ligand 9 in order to investigate experimentally whether, as the authors propose, these residues are molecular determinants of selectivity for RIOK2 over RIOK1 and 3.

It would be interesting, as the reviewer suggests, to generate mutants of RIOK2 and measure the binding affinities of the various inhibitors and compare these to their affinities against RIOK1 and RIOK3 however this would be a large study and we think outside the scope of this work.

Referee: 2

Comments to the Author(s)

In the paper 'Crystal structure of human RIOK2 bound to a specific inhibitor' submitted by Jing Wang et al. The authors report the first crystal structure of an inhibitor to the atypical kinase RIOK2. Here the authors implement protein purification, crystallisation, X-ray structural determination, ITC, fluorescence based cellular assay, and molecular dynamics. The authors present a suitable body of work, that is well designed and executed. The MS is also well written and a nice light read. The study is of interest to the kinase/small molecule community

We thank the reviewer for the positive evaluation and address the individual remarks below.

Introduction

Line 77/78 – use of 'Rio2' I assume this is how RIOK2 is referred to in (18)?

Yes, the equivalent yeast enzyme is known as Rio2.

Results

Line 95 – remove the word 'protein's'

We have removed the word.

Line 153 – 'Rio2' usage again

This line refers again to the yeast Rio2.

Fig.2 – In C & D could you rotate the chem draw pyridine to mimic its binding mode, i.e both N on same side (I know it is small thing just looks neater) as you have drawn it in Fig.4

We have made this change to include the compounds drawn in the same orientation as Fig. 4.

Discussion

Line 239 – ‘The previously determined dissociation constants for this inhibitor series.....’ I am mildly confused with this and the origins of the SAR (Varin et al) when I read the paper in absence of referring to the Varin et al work. How I review this is that the SAR are both against RIOK2, the reported Kd in this paper for compounds 9 and 10 are determined via ITC. But the Varin et al SAR is not clear the method used to obtain Kd’s in this paper. Need to reference the KdELECT DiscoverX displacement method here, otherwise, hard to compare the 393 and 349nM reported from ITC for compounds 9 and 10 against Fig.4 with 520 and 160nM for 9 and 10, respectively. I have no problem with the measurements and the authors interpretations, I would just like a little clarity of the origins.

We have modified this sentence to say “The previously determined dissociation constants for this inhibitor series [22] using the DiscoverX KdELECT measurement method....”